# Temporal trends in overweight and obesity and chronic disease risks among adolescents and young adults: A ten-year review at a tertiary institution in Nigeria

**Abayomi Olabayo Oluwasanu**[1]*, **Joshua Odunayo Akinyemi**[2], **Mojisola Morenike Oluwasanu**[3], **Olabisi Bada Oseghe**[1], **Olusola Lanre Oladoyinbo**[1], **Jelili Bello**[1], **Ademola Johnson Ajuwon**[3], **Ayodele Samuel Jegede**[4], **Goodarz Danaei**[5], **Olufemi Akingbola**[1]

**1** University Health Services, University of Ibadan, Ibadan, Nigeria, **2** Department of Epidemiology and Medical Statistics, Faculty of Public Health, College of Medicine, University of Ibadan, Ibadan, Nigeria, **3** Department of Health Promotion and Education, Faculty of Public Health, College of Medicine University of Ibadan, Ibadan, Nigeria, **4** Department of Sociology, Faculty of the Social Sciences, University of Ibadan, Ibadan, Nigeria, **5** Department of Global Health and Population, Harvard T. Chan School of Public Health, Harvard University, Boston, Massachusetts, United States of America

* bayofayemi2002@yahoo.com, bayooluwasanu21@gmail.com

**Data Availability Statement:** The study is a retrospective review of medical records with no contact with human subjects. The data analysed in

## Abstract

There is an increasing prevalence of obesity among college/university students in low- and middle-income countries, similar to the trend observed in high-income countries. This study aimed to describe the trend and burden of overweight/obesity and emerging associated chronic disease risks among students at the University of Ibadan (UI), Nigeria. This is a ten-year retrospective review of medical records of students (undergraduate and post-graduate) admitted between 2009 and 2018 at UI. Records of 60,168 participants were analysed. The Body Mass Index (BMI) categories were determined according to WHO standard definitions, and blood pressure was classified according to the Seventh Report of the Joint National Committee on Prevention, Detection, Evaluation and Treatment of High Blood Pressure (JNC7). The mean age of the participants was 24.8, SD 8.4 years. The majority were ≤ 40 years (95.1%). There was a slight male preponderance (51.5%) with a male-to-female ratio of 1.1:1; undergraduate students constituted 51.9%. The prevalence of underweight, over-weight, and obesity were 10.5%, 18.7% and 7.2%, respectively. We found a significant association between overweight/obesity and older age, being female and undergoing post-graduate study (p = 0.001). Furthermore, females had a higher burden of coexisting abnormal BMI characterised by underweight (11.7%), overweight (20.2%) and obese (10.4%). Hypertension was the most prevalent obesity-associated non-communicable disease in the study population, with a prevalence of 8.1%. Also, a third of the study population (35.1%) had prehypertension. Hypertension was significantly associated with older age, male sex, overweight/obesity and family history of hypertension (p = 0.001). This study identified a higher prevalence of overweight and obesity than underweight among the participants, a double burden of malnutrition and the emergence of non-communicable disease risks with potential lifelong implications on their health and the healthcare system. To address these

the study is from a third party, the University Health Services (UHS), University of Ibadan and usage was approved by the Social Sciences and Humanities Research Ethics Committee (SSHREC), University of Ibadan, Nigeria [UI/SSHEC/2020/0021]. Data cannot be shared publicly because of the legal and ethical restrictions involving hospital records and patient information. However, data may be made available with the permission of UHS management through the Director (contact: dir_uhs@mail1.ui.edu.ng, ronkeajav@yahoo.com) following a reasonable request by researchers who meet the criteria for access to confidential data after appropriate protocol submission to SSHREC (contact via: as.jegede@mail.ui.edu.ng, sayjegede@gmail.com, referring to the UI/SSHEC/2020/0021).

**Funding:** AOO received financial support for capacity building and the conduct of this study from the office of the 12th Vice-Chancellor, University of Ibadan, Nigeria. None of the co-authors received any financial benefit for the conduct of the study. The Management of the University of Ibadan had no role in study design, data collection and analysis, decision to publish, or manuscript preparation.

**Competing interests:** The authors have declared that no competing interests exist.

issues, cost-effective interventions are urgently needed at secondary and tertiary-level educational institutions.

## Introduction

Over the past two decades, dramatic increases have occurred globally in the prevalence of obesity in both children and adults [1, 2]. The prevalence of obesity in children and young adults has steadily increased to an epidemic proportion in high-income nations. The same trend is being observed in Low- and Middle-Income Countries (LMIC) [3–5]. The increasing trend of overweight and obesity among children and young adults, especially college/university students, is becoming alarming. In the USA and UK, the prevalence of being overweight or obese among young adults ranges from 22 to 35% [6–8]. Similarly, among university students in LMIC, the prevalence of overweight/obesity is reported to be 10–20.7% in Nigeria [9, 10], 10.8–24% in South Africa [11, 12], 11–37.5% in India [13, 14] and 20–30% in Malaysia [15, 16].

Of great concern is that once established, childhood and adolescent obesity status conferred markedly heightened risks for overweight and obesity in adulthood [17, 18]. The rising prevalence of overweight and obesity and lack of physical activity contributes to increased risks of various chronic diseases in young adults with greater severity in adulthood. Studies show that risk factors for metabolic syndrome are more prevalent among overweight or obese children and young adults than among those with healthy weight [19, 20]. Obesity in childhood or adolescence is associated with a higher risk of adult hypertension, coronary heart disease, and stroke [21]. Thus, overweight and obesity in adolescents and young adults have important public health implications, not just about its increasing prevalence but because of possible long-term associations with future weight status and related morbidity. There is a concern that overweight/obesity and associated chronic diseases such as hypertension, cardiovascular disease (CVD) and diabetes are fast emerging as the most prevalent non-communicable diseases (NCDs) in LMIC, prematurely affecting adolescents and younger adults. Also, while diseases associated with undernutrition are still a major issue, LMICs are experiencing a marked increase in overweight and obesity associated with the rising burden of NCDs.

The causes of excess weight gain in the young are similar to those in adults. However, the young are significantly prone to obesity due to changes during the transition from childhood/adolescence to adulthood [22], accompanied by the peculiar and significant lifestyle changes that occur at the time of leaving home for university or college education [23, 24]. The interaction of social, psychological and biological factors during these transition years, added to the pervading obesogenic environment, making them vulnerable to many risk-taking behaviours [25–27]. The transition to independence of college students, the competing academic demands in the presence of unhealthful lifestyle options and existing environments that greatly favour high energy intake and low energy expenditure provide the complex mix that may perpetuate the obesity trajectory on the campuses [28].

The period of stay of adolescents and young adults in the university offers many opportunities for relevant longitudinal studies to guide prevention and treatment services regarding obesity and associated chronic diseases. However, adolescents and young adults are overlooked mainly due to the perception that they are at low risk of developing chronic diseases. There are limited data to show the temporal trends in the transition pattern and prevalence of overweight/obesity among university students in Nigeria and other countries in Africa.

Furthermore, unlike substance abuse and mental health issues, many college and university leaders view helping overweight and obese students as outside the purview of higher education [28]. All these may explain why there have not been any recorded obesity prevention interventions in developing countries for this age group [29]. This study aimed to identify and describe the burden of overweight/obesity and chronic disease risks among adolescents and young adults at the University of Ibadan (UI), Nigeria, in order to lay the foundation for a sustainable plan that promotes optimal weight management and future research in obesity prevention and intervention in the institution.

## Materials and methods

### Study design

The study was a ten-year retrospective review of medical records of students (undergraduate and post-graduate) admitted into UI between 2009 and 2018. The study participants consisted of adolescents and young adults enrolled as freshmen for undergraduate studies and those who had returned or were enrolled for postgraduate studies in the institution. The medical record review utilised the data of students admitted for each new academic year under review. It consisted of data from the routine medical screening done at inception as part of their admission processes and the follow-up information on the treatment and care received for any obesity-related diseases at the University Health Services (UHS). In this study, adolescents and young adults were classified as 16 to 40 years of age, while those above 40 years and $\leq$ 65 years were classified as adults (middle age and older adults).

### Study setting

UI was established in 1948 and is located five miles (8 kilometres) from the centre of Ibadan city in Southwestern Nigeria. It runs academic programmes in sixteen Faculties and other academic units, including the Institutes of Child Health, Education, and African Studies. UI is the premier university and the flagship of postgraduate study in Nigeria. It has a unique status and recruitment policy that covers and offers admission to eligible students from all the states and regions of Nigeria. Also, UI is host to the Pan African University (PAU) which is a Continental initiative of the African Union Commission (AUC). PAU commenced officially in 2011 and has five Institutes located in the five sub-regions of Africa. Pan African University for Life and Earth Sciences Institute (including Health and Agriculture), PAULESI is located in UI and is designed to run only postgraduate programmes.

### Study population

A total sample consisting of all the medical records of students during the period in review was extracted. Specifically, all the health records of the newly admitted undergraduate (from secondary schools in Nigeria) and postgraduate students during the ten years of review (2009–2018) were used for this study. Some of the postgraduate students were former UI students who had completed their undergraduate studies and returned for postgraduate admission and had medical screening repeated during the period of this study. Also, the study participants included postgraduate students admitted from other tertiary institutions in Nigeria and those from member states of the African Union through the Pan-African University for Life and Earth Sciences Institute (including Health and Agriculture), PAULESI. UI offered admission to an average of 3,500 undergraduate students from secondary schools and 3,000 postgraduate students yearly during the period in review. Therefore, a total sample of approximately 60,168 medical records with complete information was used for this study.

## Data collection

Each academic year, new students admitted for undergraduate and postgraduate programmes are required to register at UHS as part of the admission process and for their medical care during their stay at UI. The UHS employs structured questionnaires to collect pre-admission health information, including the students' socio-demographic characteristics and background medical (personal and family) history. Physical examination and measurements including pulse rate, blood pressure (BP), height, weight and BMI were also carried out. Weight was measured to the nearest 0.1 kg using a digital scale. Height was measured to the nearest 0.1 cm with a portable stadiometer while the individual stood barefoot on the centre of the base with their back to the stadiometer. The BMI was calculated as body weight (kg) divided by squared height (m2). The BMI was then classified into four categories: underweight (BMI < 18.5), normal weight (BMI 18.5–24.9), overweight (BMI 25.0–29.9) and obesity (BMI $\geq$ 30) [30]. Blood pressure was measured using a mercury sphygmomanometer based on the recommended standards [31] and classified according to JNC 7 [32]. Chest x-ray and urinalysis were carried out as part of the initial medical evaluation. All these and the records of subsequent visits by each student to the clinic were documented in their medical files. Information about diagnosis and treatment for obesity-related diseases such as hypertension, diabetes, dyslipidemia, asthma, and gall stones were extracted and reviewed retrospectively. Also, the results of relevant investigations such as Electrocardiography (ECG) and lipid profile were reviewed, where available.

The medical records were extracted by trained Research assistants using a standardised excel tool from June to October 2020. The outcome variables of interest were underweight, overweight and obesity, and the covariates were age, sex, background medical (personal and family) history of the students (hypertension, diabetes, asthma, medications), treatment records for obesity-related diseases such as *hypertension*, *diabetes*, *dyslipidemia*, *asthma*, *gall stones and cancers*.

## Data analysis

The data were entered and cleaned using SPSS 21. Numerical variables such as age, weight, height and blood pressure were summarised using mean (SD), while frequencies/percentages were used for categorical variables. The prevalence of underweight, overweight and obesity and their trends during the period in review were determined. The association between the students' sociodemographic characteristics, hypertension and overweight/obesity was assessed using Chi-square ($\chi^2$) at a 5% significance level. The factors associated with obesity/overweight and hypertension were evaluated using a multivariable binary logit model. Measures of association were reported as Odds Ratio with a 95% Confidence Interval (95% CI).

## Ethical consideration

This is a retrospective medical record review only with no contact with human subjects. Permission to access the data was obtained from the Management of the University of Ibadan Health Services. The Social Sciences and Humanities Research Ethics Committee (SSHREC), University of Ibadan, Nigeria, provided ethical approval for this study [UI/SSHEC/2020/0021]. The study was conducted following the National Code of Health Research Ethics, Nigeria, in accordance with the Declaration of Helsinki guidelines. All data were fully anonymized before accessing them. The information extracted from the medical records was recorded without identifiers and kept confidential. The SSHREC waived the need for informed consent for the study.

**Table 1. Socio-demographic characteristics of students in each year of entry at the University of Ibadan from 2009–2018.**

| Variables | Overall (%) | 2009 (%) | 2010 (%) | 2011 (%) | 2012 (%) | 2013 (%) | 2014 (%) | 2015 (%) | 2016 (%) | 2017 (%) | 2018 (%) |
|---|---|---|---|---|---|---|---|---|---|---|---|
| | n = 59732 | n = 5739 | n = 6726 | n = 6630 | n = 7288 | n = 5949 | n = 5146 | n = 6314 | n = 5259 | n = 4860 | n = 5821 |
| **Age-group** (years) | | | | | | | | | | | |
| 16–20 | 23460 (39.3) | 2002 (34.9) | 2234 (33.1) | 1839 (27.8) | 1817 (24.9) | 2084 (34.9) | 2303 (43.9) | 2693 (42.8) | 2621 (49.8) | 2741 (56.4) | 3126 (53.7) |
| 21–25 | 14646 (24.5) | 1359 (23.7) | 1446 (21.5) | 1151 (17.4) | 1630 (22.4) | 1693 (28.4) | 1272 (24.7) | 1979 (31.3) | 1476 (28.1) | 1255 (25.8) | 1385 (23.8) |
| 26–30 | 11460 (19.2) | 1083 (18.8) | 1512 (22.5) | 1842 (27.7) | 1965 (26.9) | 1213 (20.5) | 870 (16.9) | 975 (15.4) | 731 (13.9) | 494 (10.2) | 775 (13.3) |
| 31–35 | 4687 (7.9) | 576 (10.1) | 645 (9.6) | 836 (12.6) | 834 (11.4) | 455 (7.7) | 358 (7.0) | 342 (5.4) | 206 (3.9) | 161 (3.3) | 274 (4.7) |
| 36–40 | 2625 (4.4) | 366 (6.4) | 442 (6.6) | (436) 6.6 | 489 (6.7) | 240 (4.1) | 165 (3.2) | 155 (2.5) | 117 (2.2) | 102 (2.1) | 113 (1.9) |
| 41+ | 2854 (4.8) | 353 (6.2) | 447 (6.7) | 526 (6.7) | 553 (7.6) | 264 (4.4) | 178 (3.5) | 170 (2.7) | 108 (2.1) | 107 (2.2) | 148 (2.5) |
| **Sex** | | | | | | | | | | | |
| Male | 30768 (51.5) | 3011 (52.5) | 3597 (53.5) | 3563 (53.7) | 4057 (55.7) | 3106 (52.2) | 2625 (51.0) | 3146 (49.8) | 2410 (45.8) | 2394 (49.3) | 2859 (49.1) |
| Female | 28964 (48.5) | 2728 (47.5) | 3129 (45.5) | 3067 (46.3) | 3231 (44.3) | 2843 (47.8) | 2521 (49.0) | 3168 (50.2) | 2849 (54.2) | 2466 (50.7) | 2962 (50.9) |
| **Programme** | | | | | | | | | | | |
| Undergraduate | 31022 (51.9) | 3057 (53.3) | 3289 (48.8) | 2369 (35.7) | 2437 (33.6) | 2969 (49.9) | 3049 (59.3) | 3500 (55.4) | 3220 (61.2) | 3312 (68.2) | 3820 (65.6) |
| Postgraduates | 28710 (48.1) | 2682 (46.7) | 3437 (51.1) | 4261 (64.3) | 4851 (66.6) | 2980 (50.1) | 2097 (40.8) | 2814 (44.6) | 2039 (38.8) | 1548 (31.8) | 2001 (34.4) |

# Results

## Background characteristics

Table 1 shows the trends in socio-demographic characteristics (age group, sex, programme mode (undergraduate or postgraduate), BMI and hypertension) of students between 2009 and 2018. The majority were ≤ 40 years (95.1%). Students aged 16–20 years constituted over a third of the study population. In addition, the percentage of participants aged 16–20 years increased steadily from 34.9% in 2013 to 53.8% in 2018, while those aged 36–40 years declined from 4.1% to 1.9%. The sex distribution favoured males between 2009 (51.5%) and 2011 (53.7%), but this was reversed such that males constituted less than 50% since 2015. Overall, undergraduate students constituted 51.9% of the entire sample. However, there were some variations over time, with the lowest value in 2011 (35.8%) and the highest in 2017 (68.1%) and 2018 (65.8%).

## Anthropometrics and hypertension profile

Overall, 10.5% of students were underweight [see Table 2]. On the one hand, there was a rising trend in the percentage of participants who were underweight between 2009 and 2017 [Table 2]. The prevalence of overweight and obesity was 18.7% and 7.2%, respectively (Table 2). Over time, the overweight prevalence ranged between 23.0% in 2009 and 16.2% in

**Table 2. BMI and hypertension prevalence at each year of entry of students admitted to the University of Ibadan 2009–2018.**

| Variables | Overall (%) | 2009 n (%) | 2010 n (%) | 2011 n (%) | 2012 n (%) | 2013 n (%) | 2014 n (%) | 2015 n (%) | 2016 n (%) | 2017 n (%) | 2018 n (%) |
|---|---|---|---|---|---|---|---|---|---|---|---|
| **BMI** | | | | | | | | | | | |
| Underweight | 6311 (10.5) | 331 (5.8) | 584 (8.7) | 598 (9.0) | 563 (7.7) | 659 (11.1) | 692 (13.2) | 905 (14.2) | 627 (11.9) | 751 (15.5) | 619 (10.6) |
| Normal | 38038 (63.5) | 3527 (62.1) | 4339 (64.4) | 3919 (59.1) | 4468 (61.2) | 3953 (66.0) | 3466 (66.2) | 3866 (60.7) | 3581 (68.2) | 3240 (66.7) | 3877 (66.5) |
| Overweight | 11186 (18.7) | 1304 (23.0) | 1313 (19.4) | 1464 (22.1) | 1599 (21.9) | 993 (16.7) | 817 (15.6) | 1127 (17.7) | 772 (14.7) | 652 (13.4) | 945 (16.2) |
| Obese | 4339 (7.2) | 517 (9.1) | 506 (7.5) | 649 (9.8) | 672 (9.2) | 375 (6.3) | 260 (5.0) | 470 (7.4) | 272 (5.2) | 216 (4.4) | 385 (6.6) |
| Mean ±SD | | 23.76 ±4.40 | 23.20 ±4.40 | 23.61 ±4.65 | 23.61 ±4.41 | 22.70 ±4.19 | 22.34 ±4.09 | 22.34 ±6.07 | 22.42 ±5.32 | 21.94 ±3.99 | 22.68 ±4.40 |
| **Hypertension** | | | | | | | | | | | |
| Yes | 4851 (8.1) | 476 (8.3) | 556 (8.3) | 624 (9.4) | 419 (5.7) | 483 (8.1) | 441 (8.6) | 723 (11.5) | 444 (8.5) | 362 (7.4) | 323 (5.5) |
| No | 54894 (91.9) | 5273 (91.7) | 6170 (91.7) | 6005 (90.6) | 6874 (94.3) | 5465 (91.9) | 4706 (91.4) | 5589 (88.5) | 4809 (91.5) | 4503 (92.6) | 5498 (94.5) |

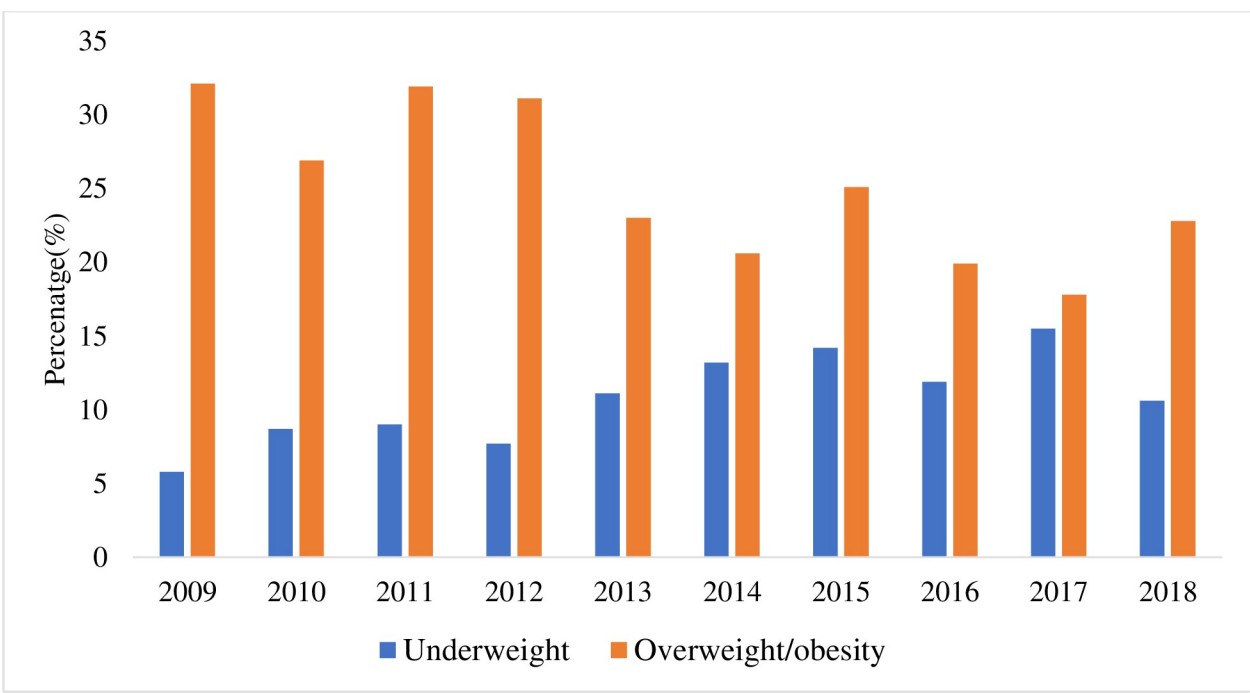

**Fig 1. Trends in the prevalence of underweight and overweight/obesity (overnutrition) across each year of admission between 2009 and 2018.**

2018. Similarly, the percentage of obese individuals ranged between 9.1% in 2009 and 6.6% in 2018 [Table 2]. There was a dip in the percentages of both overweight and obese students in 2014, 2016 and 2017. On the other hand, except for 2017, there was a consistently higher prevalence of overweight than underweight in each year of admission, and a shift from underweight to overnutrition (overweight and obesity) across each year of admission into the University [Table 2, Fig 1]. Regarding sex distribution, there were more underweight, overweight, and obese females than males [Fig 2].

The prevalence of hypertension based on systolic (140) and diastolic (90) blood pressure cut-off at enrolment was 8.1%. The level seems to be on a downward trend from 2015 (11.4%) to 2018 (5.5%) [Table 2].

### Medical history and obesity-related health conditions

Table 3 shows the students' medical history and obesity-related health conditions or disease profiles. Over the ten years, hypertension was the most prevalent non-communicable disease, followed by asthma. Thirty-five per cent of the study population had prehypertension, defined as systolic pressure from 120 to 139mmHg or a diastolic pressure from 80 to 89mmHg. About 2.2% of the study population had multiple risk factors for hypertension. The multiple risk factors identified in the data include age, male sex, family history of hypertension, overweight and obesity. Other rare but associated diseases include diabetes, dyslipidemia, osteoarthritis and gallstone. There was no record of cancer diagnosis. The result shows that 5.6% of those diagnosed with hypertension had Electrocardiography (ECG) done as part of their evaluation. About 75% of the ECG results revealed Left Ventricular Hypertrophy (LVH). In terms of intervention, only about a tenth (10.9%) of those who met the criteria for a hypertension diagnosis had follow-up treatment records. On the other hand, almost all the study participants diagnosed with asthma (97.9%) and diabetes (91.7%) had follow-up treatment records.

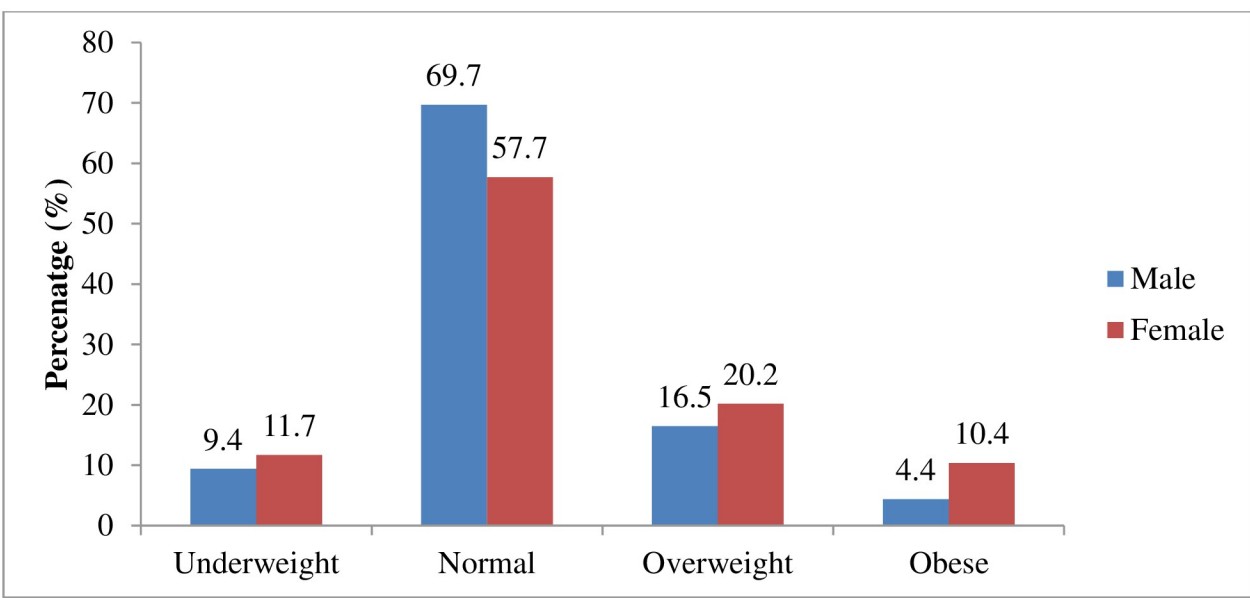

**Fig 2. BMI categories according to the sex of students in the University of Ibadan.**

## Association between the socio-demographic characteristics of the students and overweight/obesity

There was a significant association between age and overweight/obesity (Table 4). The percentage of students that were overweight or obese increased as age increased. There was also a significant association between gender and being overweight or obese, with more female students being overweight or obese (30.9%) than male students (21.2%). More postgraduate students were overweight or obese (37.8%) than undergraduates (15.0%).

The factors identified to be significantly related to overweight/obesity were also subjected to multivariate analysis, and the results are presented in Table 5. The likelihood of overweight/obesity increased with age. For instance, students aged 36–40 years (OR = 5.27) and 41 years and above (OR = 8.96) were more likely than those aged 16.-20 years to be overweight/obese. The odds of being overweight/obese were significantly higher among females than males and postgraduate students [Table 5].

## Association between the socio-demographic characteristics and prehypertension and hypertension

Table 6 shows the bivariate association between students' socio-demographic characteristics, prehypertension and hypertension. There was a significant association between age, sex and prehypertension (p-value <0.05). Furthermore, there was a statistically significant association (p-value <0.05) between body weight (overweight and obesity) and prehypertension, as a higher percentage of students that were overweight and obese, had prehypertension (45.9%) compared to normal and underweight students (35.6%).

Similar findings were observed for hypertension. Over the ten years, there was a significant association between age and hypertension (p-value<0.05). The percentage of students with hypertension increased with age such that those over 41 years had the most cases of hypertension (29.0%), unlike those aged 16–20 years (5.0%). Male students were also more hypertensive (11.2%) compared to their female counterparts (4.8%), and this difference was statistically

**Table 3. Medical history and obesity-related disease profile of students attending the University of Ibadan health centre from 2009–2018.**

| Variables | Frequency | Percentage |
|---|---|---|
| **Family History of Hypertension** | | |
| Yes | 3416 | 4.3 |
| No | 56187 | 95.7 |
| **Family History of Diabetes** | | |
| Yes | 2548 | 4.3 |
| No | 57051 | 95.7 |
| **Hypertension** | | |
| Cases at entry | 4851 | 96.4 |
| New cases at the clinic visit | 183 | 3.6 |
| **Hypertension cases on follow-up treatment (n = 5034)** | | |
| Yes | 546 | 10.9 |
| No | 4488 | 89.1 |
| **Diabetes** | | |
| Yes | 48 | 0.1 |
| No | 60115 | 99.9 |
| **Diabetes cases on follow-up treatment (n = 48)** | | |
| Yes | 44 | 91.7 |
| No | 4 | 8.3 |
| **Dyslipidemia** | | |
| Yes | 7 | 0.0 |
| No | 60156 | 100.0 |
| **Asthma** | | |
| Yes | 432 | 0.7 |
| No | 59530 | 99.8 |
| **Asthma cases on follow-up treatment (n = 432)** | | |
| Yes | 423 | 97.9 |
| No | 9 | 2.1 |
| **Gallstone** | | |
| Yes | 6 | 0.0 |
| No | 60157 | 100.0 |
| **Osteoarthritis** | | |
| Yes | 26 | 0.0 |
| No | 60137 | 100.0 |
| **Pre-hypertension (denominator excludes hypertensives)** | 20959 | 35.1 |
| **Multiple HTN risk factors** | 1315 | 2.2 |
| **Chest x-ray** | | |
| Normal | 59430 | 99.5 |
| Abnormal | 302 | 0.5 |
| **ECG (n = 282, 5.6%)** | | |
| Normal | 72 | 25.5 |
| Abnormal (LVH) | 210 | 74.5 |

significant (p-value<0.05). More postgraduate students (10.4%) had hypertension compared to undergraduate students (6.0%), and the difference was statistically significant (p-value<0.05). There is a significant association (p-value <0.05) between body weight (overweight and obesity) and hypertension, as a higher percentage of students that were overweight and obese, were hypertensive (14.2%) compared to normal and underweight students (5.9%).

**Table 4. Bivariate analysis showing the association between socio-demographic characteristics of students and overweight & obesity.**

| Variables | Overweight & Obese | | $\chi^2$ | p-value |
|---|---|---|---|---|
| | Yes (%) | No (%) | | |
| **Age-group** | | | | |
| 16–20 years | 3298 (14.0) | 20244 (86.0) | | |
| 21–25 years | 3278 (22.3) | 11406 (77.7) | 6321.219 | 0.001 |
| 26–30 years | 3582 (31.2) | 7900 (68.8) | | |
| 31–35 years | 2039 (43.6) | 2635 (56.4) | | |
| 36–40 years | 1437 (54.8) | 1187 (45.2) | | |
| 41+ | 1884 (66.4) | 955 (33.6) | | |
| **Sex** | | | | |
| Male | 6545 (21.2) | 24282 (78.8) | 730.243 | 0.001 |
| Female | 8980 (30.6) | 20067 (69.4) | | |
| **Programme** | | | | |
| Undergraduate | 4674 (15.0) | 26458 (85.0) | 4023.587 | 0.001 |
| Postgraduates | 10851 (37.8) | 17891 (62.2) | | |

The odds of prehypertension increased with age; students aged 41 years and above were three times more likely to have prehypertension than those aged 16–20 years (OR = 2.17, CI:1.94–2.43), and male students were more likely to have prehypertensive compared to females (OR = 2.40, CI:2.34–2.52).

Similarly, the odds of hypertension increased with age such that students aged 41 years and above were seven times as likely as those aged 16–20 years to have hypertension (OR = 6.11, CI: 5.29–7.06). Female students were less likely to be hypertensive (OR = 0.38, CI: 0.36–0.42) than their male counterparts. Overweight/obesity was also associated with a higher risk of hypertension (OR = 2.25, CI: 2.10–2.41) [Table 7].

## Discussion

This study identified the burden of overweight and obesity and associated risk factors for chronic, non-communicable diseases among adolescents, young and older adults. The data depicts the profile of a large population of students admitted into UI from secondary schools

**Table 5. Multivariate analysis showing the association between socio-demographic characteristics of students with overweight and obesity.**

| Variables | Odd-ratio | p-value | 95%Confidence Interval (Odd ratio) | |
|---|---|---|---|---|
| | | | Lower | Upper |
| **Age-group** | | | | |
| 16–20 years | ref | | | |
| 21–25 years | 1.30 | 0.001 | 1.21 | 1.39 |
| 26–30 years | 1.94 | 0.001 | 1.78 | 2.11 |
| 31–35 years | 3.39 | 0.001 | 3.08 | 3.74 |
| 36–40 years | 5.27 | 0.001 | 4.72 | 5.89 |
| 41+ | 8.96 | 0.001 | 8.01 | 10.02 |
| **Sex** | | | | |
| Male | ref | | | |
| Female | 2.12 | 0.001 | 2.04 | 2.21 |
| **Programme** | | | | |
| Undergraduate | ref | | | |
| Post-graduate | 1.63 | 0.001 | 1.52 | 1.75 |

**Table 6. Bivariate analysis showing the association between socio-demographic characteristics of students and pre-hypertension and hypertension.**

| Variables | Pre-hypertension | | $\chi^2$ | p-value | Hypertension | | $\chi^2$ | p-value |
|---|---|---|---|---|---|---|---|---|
| | Yes (%) | No (%) | | | Yes (%) | No (%) | | |
| **Age-group** | | | | | | | | |
| 16–20 years | 7945 (35.7) | 14312 (64.3) | 449.17 | 0.001 | 1169 (5.0) | 22285 (95.0) | 2391.217 | 0.001 |
| 21–25 years | 5118 (37.4) | 8558 (62.6) | | | 942 (6.4) | 13698 (93.6) | | |
| 26–30 years | 3922 (37.3) | 6584 (62.7) | | | 925 (8.1) | 10534 (91.9) | | |
| 31–35 years | 1702 (41.2) | 2430 (58.8) | | | 534 (11.4) | 4151 (88.6) | | |
| 36–40 years | 1042 (48.6) | 1103 (51.4) | | | 451 (17.2) | 2173 (82.8) | | |
| 41+ | 1116 (56.1) | 874 (43.9) | | | 828 (29.0) | 2026 (71.0) | | |
| **Sex** | | | | | | | | |
| Male | 13007 (47.8) | 14198 (52.2) | 2200.2 | 0.001 | 3457 (11.2) | 27313 (88.8) | 825.462 | 0.001 |
| Female | 7838 (28.5) | 19663 (71.5) | | | 1394 (4.8) | 27581 (95.2) | | |
| **Programme** | | | | | | | | |
| Undergraduate | 11014 (37.9) | 18088 (62.1) | 1.748 | 0.186 | 1861 (6.0) | 29174 (94.0) | 396.201 | 0.001 |
| Postgraduates | 9831 (38.4) | 15773 (61.6) | | | 2990 (10.4) | 25720 (89.6) | | |
| **Overweight & Obese** | | | | | | | | |
| Yes | 6043 (45.9) | 7114 (54.1) | 454.05 | 0.001 | 2193 (14.2) | 13244 (85.8) | 1056.767 | 0.001 |
| No | 14724 (35.6) | 26665 (64.4) | | | 2613 (5.9) | 41489 (94.1) | | |
| **Family history of HTN** | | | | | | | | |
| Yes | 1174 (40.2) | 1748 (59.8) | 5.946 | 0.015 | 464 (13.7) | 2922 (86.3) | 130.36 | 0.001 |
| No | 19449 (37.9) | 31832 (62.1) | | | 4516 (8.1) | 51281 (91.9) | | |

**Table 7. Multivariate analysis showing the association between socio-demographic characteristics of students and prehypertension and hypertension.**

| | Prehypertension | | | | Hypertension | | | |
|---|---|---|---|---|---|---|---|---|
| | Odds-ratio | p-value | 95%Confidence Interval (Odd ratio) | | Odds-ratio | p-value | 95%Confidence Interval (Odd ratio) | |
| | | | Lower | Upper | | | Lower | Upper |
| **Age-group** | | | | | | | | |
| 16–20 years | Ref | | | | Ref | | | |
| 21–25 years | 1.24 | 0.001 | 1.18 | 1.32 | 1.42 | 0.001 | 1.28 | 1.57 |
| 26–30 years | 1.19 | 0.001 | 1.10 | 1.28 | 1.68 | 0.001 | 1.648 | 1.91 |
| 31–35 years | 1.30 | 0.001 | 1.18 | 1.42 | 2.21 | 0.001 | 1.91 | 2.56 |
| 36–40 years | 1.73 | 0.001 | 1.55 | 1.93 | 3.38 | 0.001 | 2.89 | 3.95 |
| 41+ | 2.17 | 0.001 | 1.94 | 2.43 | 6.11 | 0.001 | 5.29 | 7.06 |
| **Sex** | | | | | | | | |
| Male | 2.40 | 0.001 | 2.34 | 2.52 | 2.53 | 0.001 | 2.37 | 2.71 |
| Female | Ref | | | | Ref | | | |
| **Programme** | | | | | | | | |
| Undergraduate | Ref | | | | Ref | | | |
| Postgraduates | 0.72 | 0.001 | 0.68 | 0.76 | 0.75 | 0.001 | 0.67 | 0.83 |
| **Overweight &Obesity** | | | | | | | | |
| No | Ref | | | | Ref | | | |
| Yes | 1.75 | 0.001 | 1.67 | 1.83 | 2.25 | 0.001 | 2.10 | 2.41 |
| **Family history of HTN** | | | | | | | | |
| Yes | 1.14 | 0.001 | 1.05 | 1.23 | 1.63 | 0.001 | 1.47 | 1.83 |
| No | Ref | | | | Ref | | | |

and postgraduate students from tertiary institutions in different states and regions of Nigeria, and also from Africa. Hence the data reflects a nationally diverse population of adolescents and young adults in Nigeria and some African countries.

## The burden of overweight and obesity

In this study, the prevalence of overweight and obesity was 18.7 and 7.2%, respectively. Similar studies conducted to determine the prevalence of overweight and obesity in Nigerian universities recorded prevalence rates of 16.2% and 4.8% [33] and 25% and 11% [34], respectively. This finding aligns with the overall prevalence of overweight and obesity in a multi-centre study among LMICs, including Nigeria, which was 22% and 5.8% [12], and in Ghana, 25.8% and 5.9%, respectively [35]. In this study, overweight and obesity had significant relationships with the older age group, being female and undergoing postgraduate training.

We found a consistently increasing trend of overweight and obesity with age, which persists significantly in the consecutive admission cohorts, suggesting an increased risk of obesity till middle age and later in adulthood. The result of our study aligns with the findings from the global study of overweight and obesity in children and adults, which showed that in both developed and developing countries, the successive cohort from 1980 to 2013 tend to gain weight at all ages, and the most rapid weight gains occurred between the ages of 20 and 40 years [36]. This indicates that without any decisive policy and interventions to tackle overweight and obesity in adolescents and young adults in developing countries, including Nigeria [29], it is unlikely that the natural course of obesity will change or be different from the pattern observed elsewhere. Where exposure to an obesogenic environment is the norm, abnormal weight gain persists, and underweight students are also at risk of transitioning to overweight and obesity across the life course.

## Double burden of malnutrition

Our findings revealed a double burden of malnutrition characterised by the co-occurrence of undernutrition and overweight and obesity. While there was a rising trend in the underweight status of the participants as the year of entry increased, there was a shift from underweight to overnutrition (overweight and obesity) across each year of admission to the University. In this study, females had a higher burden of abnormal BMI with co-occurrence of underweight, overweight and obesity, similar to studies reported in other LMICs [37, 38]. The double burden of malnutrition documented in this study is similar to findings reported among students of tertiary institutions in different geopolitical regions of Nigeria [34, 39] and other LMICs [12]. The occurrence of a double burden of malnutrition, as found in this study, reflects the Nigerian population and many developing countries experiencing nutrition and socioeconomic transitions [40–42]. This strongly implies that environmental, nutritional and socioeconomic variables rather than genetic factors are likely responsible for the ongoing dramatic double burden of malnutrition in LMICs.

The double burden of malnutrition is a complex and vital phenomenon because of the relationship and biological link between the diverse forms of malnutrition beyond coexistence. The increased prevalence of underweight in this study is worrisome because low BMI may be a risk factor for CVD and all-cause mortality [43–45]. Childhood undernutrition is associated with long-term increased susceptibility to fat accumulation mostly in the central region of the body, lower fat oxidation, lower resting and postprandial energy expenditure, insulin resistance and a higher risk of diabetes, hypertension and dyslipidaemia in adulthood [46]. At the same time, undernutrition in the form of nutritional deficiencies is an important underlying risk factor for major infectious diseases and global child mortality [47, 48]. Thus, the double burden of malnutrition at a younger age is a silent driver of the double burden of infectious

and non-communicable diseases [47]. Future medical screening in UI can provide more useful data about the dynamics and effects of the dual burden of malnutrition if the diets and nutritional status of the students are explored together with additional indices of adiposity. The link between CVD risk, underweight, and other forms of malnutrition, including micronutrient deficiencies in young adults, also needs further examination.

## Obesity-related health conditions

Hypertension was the most prevalent chronic condition among the study participants. Its prevalence was 8.1%, consistent with a study conducted among Ethiopian students, where the prevalence was 7.4% [49]. Studies indicate that 90% of adolescents and young adults with hypertension have primary or essential hypertension, with no specific cause but well-defined risk factors [50–52]. In this study, hypertension had significant relationships with the older age group, being male, undergoing postgraduate study, overweight or obese, and having a family history of hypertension. As overweight and obesity rates increased among the study population, there was a parallel rise in the prevalence of hypertension across the age groups. A similar finding has been documented among students of a tertiary institution in Cameroon [53].

Hypertension is the leading cause of death globally and the most important risk factor for cardiovascular disease, stroke, and chronic kidney disease (CKD) [32, 54]. Left ventricular hypertension (LVH) is one of the early manifestations and immediate consequences of hypertension [55, 56]. Available evidence shows that adolescents and young adults with hypertension have similar target-organ damage such as LVH, microalbuminuria and carotid intimal thickness as older adults with hypertension [57]. This study shows that only 5.6% of the study participants had an Electrocardiography (ECG) done for the evaluation of hypertension-related target-organ damage. The ECG of about three-quarters of those evaluated revealed LVH. This finding has limited usefulness and must be interpreted with caution given the relatively small proportion of the participants who had ECG and that ECG is not validated for the diagnosis of LVH in young individuals [58]. However, LVH indicates an increased risk for future cardiovascular disease. Also, the presence of LVH in adolescents and young adults with high BMI has been implicated in sudden cardiac death (SCD) [59–61]. After adjusting for age and blood pressure, the body-mass index remained a strong independent predictor of left ventricular mass, ventricular wall thickness, and left ventricular internal dimension [62, 63]. In this study, however, there was no information on the likely impact of excess body weight on LV geometry and function as this could only be assessed by imaging. LVH and other target organ markers are associated with adverse cardiovascular outcomes and risks for chronic diseases [50, 52]. Unfortunately, these complications and consequences are unlikely to be clinically apparent for many years in adolescence and young adulthood. Thus, there is a need to adapt and follow recommended guidelines for investigating and managing hypertension and co-morbidities in adolescents and young adults with abnormal BMI.

Also, our study revealed that a third of the study population (35.1%) had prehypertension. Reports from many studies indicate that prehypertension is common among adolescents and young adults, with evidence of target organ damage already present [64, 65]. Prehypertension is not considered a disease category but identifies those likely to progress to stage 1 or 2 hypertension in the future [66, 67], without intervention. It is a strong predictor of hypertension and future cardiovascular disease [68, 69]. As found in this study, the significant relationship between prehypertension and being overweight and obese implies that the burden of hypertension and cardiovascular disease may increase if the obesity epidemic continues to spiral out of control.

We found that only about a tenth of those who met the criteria for a hypertension diagnosis had records of any follow-up intervention or treatment. This is discouraging but may not be

unconnected with poor documentation and challenges associated with secondary health data. Furthermore, unlike asthma and diabetes which usually manifest with acute symptoms, hypertension is largely asymptomatic, and this may also have negatively influenced the health-seeking behaviour of the study participants. Existing studies reveal that hypertension diagnosis rates are lower, treatment is often delayed, and that control is lower in young people than in adults due to multiple factors [70–72]. Untreated hypertension in adolescents and young adults increases the risk of cardiovascular events in middle age [73]. It also contributes to an earlier onset of coronary heart disease, heart failure, stroke, and transient ischemic attacks [74].

Our findings underscore four crucial points. First, the burgeoning prevalence and pattern of overweight and obesity across the age groups from different locations suggest a strong environmental/social causative factor deserving further attention as possible targets for intervention. Second, the evidence indicates the need to implement lifestyle-related interventions as part of efforts to halt the progression of obesity and prevent the emergence of chronic diseases. Third, overweight or obesity and chronic disease risks, especially cardiovascular disease, and its potential complications, were under-assessed. The resulting underestimation may further contribute to the perception of a low-risk profile for cardiovascular disease among adolescents and young adults. Fourth, greater success may be achieved in curbing malnutrition if the existing environmental, socioeconomic, and cultural factors, including gender-specific variations, are explored further for appropriate interventions. The double burden of malnutrition offers a unique and vital opportunity for integrated action on malnutrition in all its forms. According to WHO, there is an urgent need for double-duty actions, "which are interventions, programmes, and policies that have the potential to simultaneously reduce the risk and burden of under and overnutrition" [5, 63] in the education sector.

## Implications for interventions

Primary prevention remains the most realistic strategy to curb the growing burden of obesity and obesity-related health conditions among adolescents and young adults. This is pertinent considering that overweight/obesity and related health conditions documented in some adolescents and young adults in this study predate their entry to the university. Unfortunately, systematic reviews of studies have reported gaps in the number and quality of obesity prevention interventions conducted within the adolescent and young adult age groups in developing countries [75, 76]. Adolescents and young adults represent a unique age group whose views and health needs are not adequately addressed by the health management information system. For instance, there was apparent neglect of adolescents (*except for married female adolescents*) during the collection of national data on nutrition during the Nigeria Demographic Health Surveys [40]. In addition, policies and programmes to address nutrition in Nigeria remain skewed towards undernutrition and children under five years [40, 77]. To tackle obesity among Nigerian adolescents and young adults, a critical step is the collection of data on its burden, risk factors and trends. Furthermore, there is a need for a holistic, synergistic mix of population-level interventions such as educational interventions, the provision of physical activity facilities, coupled with the regulation of labelling, marketing, content and pricing of energy-dense foods and sugar-sweetened beverages, which target the obesogenic environment and requires a multi-sectoral approach [77, 78].

## Strength and limitation of the study

The strengths of this study include the large population of adolescents and young adults from diverse geographical and social backgrounds as participants. Population-based studies on obesity and emerging NCD risks are needed to build an evidence base that socially and culturally

reflects the realities in developing countries. Hitherto, studies on obesity and overweight problems in Nigerian tertiary institutions were mainly conducted on undergraduate students. However, this study included undergraduate and postgraduate students covering a broad spectrum of adolescents and young and middle-aged adults. Furthermore, the study spans a ten-year period showing the trajectory of obesity and hypertension prevalence. The findings from this study will add to the body of knowledge on the burden of obesity and overweight among university students in Nigeria and other developing countries, which will assist in planning effective health intervention programmes to reduce the heavy burden of obesity noted among university students.

The limitations of this study include a lack of data on dietary intake, physical activity, socio-economic status, smoking, alcohol consumption, and other emerging lifestyle risk factors for obesity and overweight, which could have provided a richer perspective on the factors contributing to the burden of obesity and associated diseases among this population. The study described a single-centre experience in Nigeria, the results may not apply to the general population of young adults in Africa. However, this study provides additional information on the chronic disease risks associated with higher BMI in young adults and the implication of the dual burden of malnutrition to the increasing prevalence of NCDs. It provides a large database for university managers to launch integrated public health interventions for malnutrition and NCDs at a crucial time in the life of adolescents and young adults.

## Conclusion

This study has identified the burden of double malnutrition, rising trends in the prevalence of overweight/obesity among students in a tertiary institution and the emergence of chronic disease risks with lifelong implications on their health and a concomitant burden on the health system. Evidence-based, cost-effective interventions are urgently needed at the secondary and tertiary-level educational institutions to address the growing burden of malnutrition and chronic disease risks among the adolescent and young adult population. These interventions must be holistic and transcend obesity awareness programmes to include those which target the obesogenic physical and policy environments and empower adolescents and young adults to adopt appropriate healthy behaviours.

## Acknowledgments

AOO is grateful to The Royal Tropical Institute (KIT), Amsterdam, the Netherlands, which provided training on the 'Control Strategies for Communicable and Non-communicable Diseases' through the Orange knowledge Scholarship. Also, the Bernard Lown Scholars Program, Harvard T. Chan School of Public Health, provided Fellowship training in Cardiovascular Disease Prevention and support to AOO for the conduct of this study. A special appreciation goes to the 12th Vice-Chancellor of the University of Ibadan, Prof. Abel Idowu Olayinka, who provided institutional support to AOO for capacity building and the conduct of the study.

## Author Contributions

**Conceptualization:** Abayomi Olabayo Oluwasanu.

**Data curation:** Abayomi Olabayo Oluwasanu, Joshua Odunayo Akinyemi, Mojisola Morenike Oluwasanu, Olusola Lanre Oladoyinbo.

**Formal analysis:** Abayomi Olabayo Oluwasanu, Joshua Odunayo Akinyemi, Mojisola Morenike Oluwasanu.

**Funding acquisition:** Abayomi Olabayo Oluwasanu.

**Methodology:** Abayomi Olabayo Oluwasanu, Joshua Odunayo Akinyemi, Mojisola Morenike Oluwasanu.

**Project administration:** Abayomi Olabayo Oluwasanu.

**Resources:** Abayomi Olabayo Oluwasanu.

**Supervision:** Abayomi Olabayo Oluwasanu, Ademola Johnson Ajuwon, Ayodele Samuel Jegede, Goodarz Danaei.

**Validation:** Joshua Odunayo Akinyemi, Ademola Johnson Ajuwon, Ayodele Samuel Jegede, Goodarz Danaei.

**Writing – original draft:** Abayomi Olabayo Oluwasanu.

**Writing – review & editing:** Abayomi Olabayo Oluwasanu, Joshua Odunayo Akinyemi, Mojisola Morenike Oluwasanu, Olabisi Bada Oseghe, Olusola Lanre Oladoyinbo, Jelili Bello, Ademola Johnson Ajuwon, Ayodele Samuel Jegede, Goodarz Danaei, Olufemi Akingbola.

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
