## [Decision Letter · Decision Letter 0]

6 Jun 2022

PONE-D-22-08089’Temporal Trends in Obesity and Chronic disease risks among Young Adults: a 10-year Review at a Tertiary institution, NigeriaPLOS ONE

Dear Dr. Oluwasanu,

Thank you for submitting your manuscript to PLOS ONE. After careful consideration, we feel that it has merit but does not fully meet PLOS ONE’s publication criteria as it currently stands. Therefore, we invite you to submit a revised version of the manuscript that addresses the points raised during the review process.

We look forward to receiving your revised manuscript.

Kind regards,

Blessing Akombi-Inyang, Ph.D.

Academic Editor

PLOS ONE

Journal Requirements:

2. In the ethics statement in the Methods and online submission information, please ensure that you have specified (1) whether consent was informed and (2) what type you obtained (for instance, written or verbal, and if verbal, how it was documented and witnessed). If your study included minors, state whether you obtained consent from parents or guardians. If the need for consent was waived by the ethics committee, please include this information.

“A special appreciation goes to the 12th Vice Chancellor of the University of Ibadan, Prof. Abel Idowu Olayinka, who provided funding and institutional support to AOO for capacity building and the conduct of the study.”

“The authors received no specific funding for this work.”

6. Please include your tables as part of your main manuscript and remove the individual files. Please note that supplementary tables (should remain/ be uploaded) as separate "supporting information" files.

Reviewers' comments:

Reviewer's Responses to Questions

**Comments to the Author**

1. Is the manuscript technically sound, and do the data support the conclusions?

Reviewer #1: Yes

Reviewer #2: Yes

2. Has the statistical analysis been performed appropriately and rigorously? 

Reviewer #1: Yes

Reviewer #2: Yes

3. Have the authors made all data underlying the findings in their manuscript fully available?

Reviewer #1: Yes

Reviewer #2: No

4. Is the manuscript presented in an intelligible fashion and written in standard English?

Reviewer #1: Yes

Reviewer #2: No

5. Review Comments to the Author

Reviewer #1: This is a very nice article analyzing the prevalence of obesity and its associated medical conditions in an African University.

The article states increased rates of obesity as a concern in LMIC in the starting introductory lines, however later charts/tables show a decreasing or stable trend and in fact a rising trend for underweight status. I agree that there is a double epidemic of under and over nutrition. However this cannot be generalized to the population based on the prevalence numbers of this study which is only from University students. In the general population there may be a different trend which this study is not able to predict. This needs to be mentioned in the discussions as a limitation.

I do not think we can generalize the prevalence of an obesity epidemic from the study to the general population in the African sub continent. I think this needs to be clearly mentioned as a limitation. Additionally some of the sentences in the introduction seem to suggest an obesogenic environment in the University, however I am unsure how the authors drew such a conclusion with the data from the study showing an increase in underweight students in the enrollment category. Also this article did not study a difference between the same students increasing in weight between their first and subsequent years of enrollment and hence is unable to draw on such a conclusion. I think the tone of the introduction could be limited to only mentioning the lack of prevalence data such as obesity, hypertension in male/female students and trends and the need to study this further.

It is unclear if the higher female enrollment is causing a rising trend in obesity, though less likely this needs to be noted in the discussions as a suggestion for future study and the reasons why? Is this because of pregnancy, marital status improved socioeconomic status?

LVH may not be detected in early hypertension. It is a poor metric to determine presence of hypertension or the need for treatment. I think using the JNC criteria probably is a better way of assessing presence of hypertension and need to treat it. Given that this is a study of prevalence, I would suggest not mentioning left ventricular hypertrophy as this study was not designed to identify the prevalence of this as not all patients had an ECG.

Reviewer #2: Dear author, while i congratulate you on the time and effort you have put into preparing this manuscript, I feel that there are significant changes that will improve the quality of your manuscript.

You will find below, detailed suggestions that you may find useful and decide to adopt.

I feel that these suggestions will improve the quality of your manuscript.

I will also advice that you make your work more concise so that the relevant information can be easily identified.

You manuscript will benefit from the services of an English language editor.

Title: The title appears appropriate and descriptive of the study done. However, uniformity is required in the use of lower or upper case for letters which begin each word. It appears rather haphazard at the moment. Also, please write “10” as “Ten”

Abstract:

The abstract summarizes the work done and relevant outcome. A large majority was ≤ 40 years (95.1%) Line 53: should be change to “ Majority of the students samples were ……”

Line 58: Furthermore, females had a higher burden of coexisting abnormal BMI characterised by the co occurrence of underweight, overweight and obesity”.

Please give the exact values or the three abnormal BMI parameters in women.

Line 62: Hypertension is significantly associated with age, male sex, overweight/obesity and family history of hypertension. Was any test of statistical significance done? Please state this if any was done .

Did you mean increasing age?

Key words: seem to be appropriate

Introduction

Lines 92 and 93 and in other parts of the manuscript: Replace y” Young people “ with the young”

The causes of excess weight gain in young people are similar to those in adults. However, young 93 people

Line 95: “ during leaving home.” Replace the “ at time of leaving home for university or college education.

I find the introduction rather too lengthy. It could be made more concise and much more to the point.

Study setting

Line 153: please delete “the major”

Line 179 : please change to “during the period in review”

Line 182 : please change “for each year of entry during the review 183 period.” to “…yearly for the period in review’

Line 189: Under weight should be included as a variable.

Use the abbreviation UI to represent University of Ibadan after you have used the first abbreviation

Line 216: what is meant by “programme”?

Line 217: Consider deleting “large” . Simply say “ a majority”

Line 218: “a quarter of what’? Also, delete “the years under review”…this phrase has been used too often.

Lines 218 to 220 : Please reconstruct this sentence to make the meaning clearer. “In addition, the 219 percentage in this age group has been on the increasing trend since 2013 till 2018 while the reverse 220 pattern was observed for age 36 years and above of which their percentage declined”.

Lines 233 to 234: is not well written and has been repeated. Please re write.

Line 260: Except “ in students”

Line 314: Use “UI”

Line 315: Your sample categorization does not only include young adults, but older adults to.

Lines 329 to 330: This trend persists 329 till later years of postgraduate programmes depicting increasing risk of obesity at middle age and 330 later in adulthood. ( This sentence needs to be re written to be make it clearer)

Line 338: “ without investment in the built environment for physical activity.” The meaning of this is not clear. Please clarify.

Line 347: Please what is meant by “increasing shift”? Does this mean that under nutrition was decreasing and over nutrition was increasing as the year of entry increased? Please make your sentences clearer and easier to follow and understand. You can cite any of the figures that illustrates this and give a reason for this finding.

Line 343: The discussion in the section on Double Burden of Malnutrition should reflect the findings and numbers reported in the results. This is not the case presently. I will suggest a re write of the this section discussing the figures and relevance of both under weight and over weight , comparison with other reports , causes and implications to the wider society.

403 : change tiny to the exact percentage>

433 : “Young people” should be changed to “ Adolescents and young adults” all over the manuscript as has been used previously in your manuscript.

Though this manuscript’s title focuses on Obesity, much more ought to have been said about under weight (under nutrition )

Interventions : This will include education of the harmful effect of poor nutrition and advice and on the benefits of healthy eating including components of a balanced diet. The positive effect of role models. Deterrents for higher energy drinks and policy statements such as increased tax on such high energy drinks etc

Though i could find the figures, there were no tables present for inspection.

A major flaw of this manuscript is that the author’s posture and discussion is focused on the students and their learning environment (UI). This manuscript gives the impression or idea that these are students who have been in UI. However, this is not the case. The sample used are fresh students who have been out of the school and are just resuming into UI. Therefore, this sample is not affected or do not have any impact yet by the school environment. The BMI measurements were taken at the point of admission into the school and not during their study. Therefore, this does not reflect the impact of UI environment as the authors have made it appear. This should be corrected all over the manuscript.

The other issue is that the manuscript is much too long and almost boring. The information can be given in fewer words and in clear sentences too. The authors should engage an English language editor to help make their work clearer and more focused.

I will also suggest that the tables be included in the submission.

6. PLOS authors have the option to publish the peer review history of their article (what does this mean?). If published, this will include your full peer review and any attached files.

Reviewer #1: No

Reviewer #2: No

---

## [Author Response · Author response to Decision Letter 0]

4 Aug 2022

Response to the Academic Editor:

Thank you for the guidance, the PLOS ONE’s style requirements have been incorporated accordingly in all sections.

This is a retrospective study of medical records. All data were fully anonymized before we accessed them. The ethical approval for this study was provided by the Social Sciences and Humanities Research Ethics Committee (SSHREC), University of Ibadan (UI), Nigeria [UI/SSHEC/2020/0021]. The SSHREC waved the need for informed consent for the study. Line 204-211.

This information has been included in the ethics statement. Thank you.

Including funding information in our acknowledgement is regretted. We have deleted the funding information in the Acknowledgement Section. This section has been updated and should read:

“AOO is grateful to The Royal Tropical Institute (KIT), Amsterdam, the Netherlands, through Orange knowledge Scholarship, which provided training on the ‘Control Strategies for Communicable and Non-communicable Diseases.’ Also, the Bernard Lown Scholars Program, Harvard T. Chan School of Public Health, provided Fellowship training in Cardiovascular Disease Prevention and support to AOO for the conduct of this study. A special appreciation goes to the 12th Vice-Chancellor of the University of Ibadan, Prof. Abel Idowu Olayinka, who provided institutional support to AOO for capacity building and the conduct of the study”.

Including funding information in our acknowledgement is regretted. We have deleted the funding information in the Acknowledgement Section. This section has been updated and should read:

“AOO is grateful to The Royal Tropical Institute (KIT), Amsterdam, the Netherlands, through Orange knowledge Scholarship, which provided training on the ‘Control Strategies for Communicable and Non-communicable Diseases.’ Also, the Bernard Lown Scholars Program, Harvard T. Chan School of Public Health, provided Fellowship training in Cardiovascular Disease Prevention and support to AOO for the conduct of this study. A special appreciation goes to the 12th Vice-Chancellor of the University of Ibadan, Prof. Abel Idowu Olayinka, who provided institutional support to AOO for capacity building and the conduct of the study”.

Amendments to the Acknowledgement section and Funding statement have been included in the cover letter for your kind attention.

In the online submission form we would like to update the funding statement to read:

AOO received financial support for capacity building and the conduct of this study from the office of the 12th Vice-Chancellor, University of Ibadan, Nigeria. None of the co-authors received any financial benefit for the conduct of the study. The Management of the University of Ibadan had no role in study design, data collection and analysis, decision to publish, or preparation of the manuscript.

Thank you for the guidance. The study is a retrospective review of medical records only. The data analysed in the study is from a third party, the University Health Services (UHS), University of Ibadan and usage was approved by the Social Sciences and Humanities Research Ethics Committee (SSHREC), University of Ibadan, Nigeria [UI/SSHEC/2020/0021]. Data cannot be shared publicly because of the ethical and legal restrictions involving hospital records and patient information.

However, data may be made available with the permission of UHS management through the Director (contact: dir_uhs@mail1.ui.edu.ng, ronkeajav@yahoo.com) following a reasonable request by researchers who meet the criteria for access to confidential data after appropriate protocol submission to SSHREC (contact via as.jegede@mail.ui.edu.ng, sayjegede@gmail.com, referring to the UI/SSHEC/2020/0021). 

We would appreciate it if this information could be modified to reflect the changes in our updated Data Availability statement above. Thank you.

The tables were uploaded separately and were erroneously omitted in the old manuscript PDF file. This has been updated and the tables have been included.

The manuscript has been revised and all the typographical and grammatical errors have been corrected. Thank you.

Reviewer #1:

Reviewer: This is a very nice article analyzing the prevalence of obesity and its associated medical conditions in an African University. The article states increased rates of obesity as a concern in LMIC in the starting introductory lines, however later charts/tables show a decreasing or stable trend and in fact a rising trend for underweight status. I agree that there is a double epidemic of under and over nutrition. However, this cannot be generalized to the population based on the prevalence numbers of this study which is only from university students. In the general population there may be a different trend which this study is not able to predict. This needs to be mentioned in the discussions as a limitation.

Thank you for your kind words.

We found a consistently increased trend of overweight and obesity with age (table 4) with statistical significance. Line 302, 313-315. Also, there was an increasing shift from underweight to overnutrition (overweight/obesity) across each year of entry into the University (table 2, fig 3). Line 234. This information (fig 3) was inadvertently omitted initially but has been included in the revised manuscript as guided by your comment. Thank you.

Reviewer: I do not think we can generalize the prevalence of an obesity epidemic from the study to the general population in the African subcontinent. I think this needs to be clearly mentioned as a limitation. Additionally, some of the sentences in the introduction seem to suggest an obesogenic environment in the University, however I am unsure how the authors drew such a conclusion with the data from the study showing an increase in underweight students in the enrollment category. Also, this article did not study a difference between the same students increasing in weight between their first and subsequent years of enrollment and hence is unable to draw on such a conclusion. I think the tone of the introduction could be limited to only mentioning the lack of prevalence data such as obesity, hypertension in male/female students and trends and the need to study this further.

Response: We agree that this study described a single-centre experience which may not reflect the same situation with the general population in the Africa continent, this limitation has been included in the revised manuscript. Line 567. However, our results fall in line with other findings about increasing high BMI status in adolescents and young adults in Nigeria and other developing countries in Africa.

The data in the study depicts the profile of a large population of adolescents, young and older adults who came in as fresh students from secondary schools and postgraduate students from the University of Ibadan (UI) and other tertiary institutions from different states and regions of Nigeria. The postgraduate students from UI were former undergraduates in UI who had returned for postgraduate studies and had their medical screening repeated during the period in review. UI, as the premier university in Nigeria, has a unique status and recruitment policy that covers and offers admission to eligible students from all the states and regions of Nigeria. Hence the data reflects a nationally diverse population of young persons in the country. In fact, the PAN-African University (PAU) which is hosted by UI consist of only postgraduate students from African countries, but, unfortunately, we couldn’t determine the distribution of these categories of postgraduate students (from UI, other tertiary institutions in Nigeria and those from PAU) because the data were totally anonymized before we accessed them. We have tried to update this information in our discussion. Line 157-165.

Regarding the comment on the obesogenic environment in the university, it is true we did not study the difference in the same student between the entry point and at any other time before graduating perse. However, the significant association between overweight/obesity prevalence and increasing age and postgraduate study gives room for some cautious inferences. First, some of the study participants who enrolled for postgraduate study were returning undergraduate students of UI who had their medical screening repeated for postgraduate admission during the period under review. Like many institutions, UI gives admission preference to their young students who have good academic records for postgraduate studies before considering students from other institutions. Unfortunately, we couldn’t synthesize this information in our data because of ethical restrictions (anonymized data). Whatever it was, this category of students may have been impacted by the UI school environment. Second, the result of some studies to date shows that there has not been any form of intervention for adolescents and young adults with high BMI in universities and the general population in Nigeria and other countries in Africa. Any institution where there is no decisive policy and interventions to tackle obesity could be defined as obesogenic. Therefore, both the undergraduate and postgraduate students admitted into UI and from tertiary institutions outside of UI who were previously underweight or overweight may naturally follow the life course of the obesity trajectory, all things being equal. This is one plausible way to explain the consistent and significant rise of overweight/obesity across the age groups and extending to the postgraduate years. Of course, this and others need further study as had been indicated in the discussion. Line 394-412.

Reviewer: It is unclear if the higher female enrollment is causing a rising trend in obesity, though less likely this needs to be noted in the discussions as a suggestion for future study and the reasons why? Is this because of pregnancy, marital status improved socioeconomic status?

Response: In the study, we found an increasing trend of abnormal BMI (underweight, overweight, obesity) in females. Many reasons have been advanced in literature for this gender peculiarity, especially factors bothering on marital status, socio-cultural practices, economic disparity and those promoting gender inequality. However, our data do not contain information about any of these factors and are not part of the objectives of the study. We agree that this result and the likely causes need further examination. This point has been added to the discussion. Line 522-524. 

Reviewer: LVH may not be detected in early hypertension. It is a poor metric to determine presence of hypertension or the need for treatment. I think using the JNC criteria probably is a better way of assessing presence of hypertension and need to treat it. Given that this is a study of prevalence, I would suggest not mentioning left ventricular hypertrophy as this study was not designed to identify the prevalence of this as not all patients had an ECG.

Response: Your comment about LVH is helpful. It has made us revise the manuscript more thoroughly to bring out the relevance of LVH to our findings beyond its prevalence. LVH is one of the earliest complications of sustained hypertension in children, young and older adults. Though present, it may not be detected because of under-assessment, inappropriate tools and poor health-seeking behaviour including late presentation – all or some of these are not far-fetched in our study. LVH (both eccentric and concentric types) is also a complication of overweight/obesity with or without hypertension. It is one of the associated chronic disease risks that our study elucidated. It is an independent risk factor for cardiovascular disease and a cause of sudden cardiac death in overweight and obese adolescent and young adult or athletes. In recent times, there has been an increase in the prevalence of sudden death among young people in Nigeria necessitating a call for mandatory medical screening and ECG for young people going for the mandatory 1-year National Youth Service Corps (NYSC) and preparticipation exercise programmes. Hence, LVH deserves special attention and discussion in our study. As it turned out in our study, obesity-associated chronic health risks including LVH were grossly under-assessed in the participants. Interestingly, LVH is an eminently modifiable risk factor for CVD. Line 467-492, 519-520.

We agree with your point that using JNC is adequate to assess the presence of hypertension and the need to treat it. Thank you. 

Reviewer #2:

Reviewer: Dear author, while i congratulate you on the time and effort you have put into preparing this manuscript, I feel that there are significant changes that will improve the quality of your manuscript.

You will find below, detailed suggestions that you may find useful and decide to adopt.

I feel that these suggestions will improve the quality of your manuscript.

I will also advice that you make your work more concise so that the relevant information can be easily identified.

Your manuscript will benefit from the services of an English language editor.

Response: Thank you for your kind words and detailed comments with suggestions. The manuscript is revised accordingly to improve the scientific writing and the quality of the manuscript.

Thank you for your comments and your helpful suggestions. The engagement and conciseness of the manuscript have been revised accordingly to reflect the suggested improvements. Overall, your comments and suggestions have enhanced our engagement, conciseness and the quality of our manuscript. 

Reviewer: Title: The title appears appropriate and descriptive of the study done. However, uniformity is required in the use of lower or upper case for letters which begin each word. It appears rather haphazard at the moment. Also, please write “10” as “Ten”

Response: Apologies for the typos and grammatical errors. These have been corrected to read: 

‘’Temporal Trends in Obesity and Chronic disease risks Among Adolescents and Young Adults: A Ten-year Review at a Tertiary institution in Nigeria’’

Thank you

Reviewer: The abstract summarizes the work done and relevant outcome. A large majority was ≤ 40 years (95.1%) Line 53: should be change to “ Majority of the students samples were ……”

Response: Thank you for this comment. The correction has been effected to read:

‘The majority were ≤ 40 years (95.1%)’. Line 52.

Reviewer: Line 58: Furthermore, females had a higher burden of coexisting abnormal BMI characterised by the co occurrence of underweight, overweight and obesity”.

Please give the exact values or the three abnormal BMI parameters in women.

Response: Furthermore, females had a higher burden of coexisting abnormal BMI characterised by underweight (11.7%), overweight (20.2%) and obesity (10.4%). Line 56-58.

Reviewer: Line 62: Hypertension is significantly associated with age, male sex, overweight/obesity and family history of hypertension. Was any test of statistical significance done? Please state this if any was done.

Did you mean increasing age?

Response: Yes, statistical significance was done (p < 0.001). Line 61. Please see table 6

Yes, hypertension was significantly associated with older age, male sex, overweight/obesity and family history of hypertension (p = 0.001). Line 60-61.

Reviewer: Key words: seem to be appropriate.

Response: Thank you. The keywords have been updated to include ‘’Adolescents, Young adults’’ in the title, abstract and the body of the manuscript as you have rightly suggested below.

Reviewer: Lines 92 and 93 and in other parts of the manuscript: Replace y” Young people “with the young”

The causes of excess weight gain in young people are similar to those in adults. However, young 93 people.

Response: This has been done. Line 96 and 97. Thank you.

The causes of excess weight gain in the young are similar to those in adults. However, the young are significantly prone to obesity…Line 97.

Reviewer: Line 95: “ during leaving home.” Replace the “ at time of leaving home for university or college education.

Response: lifestyle changes that occur at the time of leaving home for university or college education. Line 99.

Reviewer: I find the introduction rather too lengthy. It could be made more concise and much more to the point.

Response: The introduction has been reduced from five to four paragraphs and rephrased concisely.

Reviewer: Line 153: please delete “the major”

Response: ‘’the major’’ deleted to read ‘from the centre of Ibadan city in Southwestern Nigeria.’ Line 144

Reviewer: Line 179 : please change to “during the period in review”

Response: medical records during the period in review was collected. Line 156.

Reviewer: Line 182 : please change “for each year of entry during the review 183 period.” to “…yearly for the period in review’

Response: and 3,000 postgraduate students yearly during the period in review. Line 165

Reviewer: Line 189: Under weight should be included as a variable

Response: The outcome variables of interest were underweight, overweight and…Line 190

Reviewer: Use the abbreviation UI to represent University of Ibadan after you have used the first abbreviation.

Response: This has been done. Thank you.

Reviewer: Line 216: what is meant by “programme”?

Response: This is about the programme mode (whether undergraduate or postgraduate). Line 215.

Reviewer: Line 217: Consider deleting “large” . Simply say “ a majority”

Response: The majority of the students were ≤ 40 years (95.1%). Line 217.

Reviewer: Line 218: “a quarter of what’? Also, delete “the years under review”…this phrase has been used too often.

Resonse: Apologies for the incomplete statement, we have rephrased the statement as ‘’ Students aged 16-20 years constituted at least a quarter of the study population. The phrase ‘’the years under review’’ also deleted accordingly. Line 217.

Reviewer: Lines 218 to 220 : Please reconstruct this sentence to make the meaning clearer. “In addition, the 219 percentage in this age group has been on the increasing trend since 2013 till 2018 while the reverse 220 pattern was observed for age 36 years and above of which their percentage declined”.

Response: In addition, the percentage of participants aged 16-20 years increased steadily from 34.9% in 2013 to 53.8% in 2018, while those aged 36-40 years declined from 4.1% to 1.9%. Line 218-219.

Reviewer: Lines 233 to 234: is not well written and has been repeated. Please re write.

Response: Thank you for pointing out this error due to the insertion of another sentence. This has been corrected as:

Similarly, the percentage of obese individuals ranged between 9.2% in 2012 and 5.0% in 2014. Line 233.

Reviewer: Line 260: Except “ in students”

Response: The sentence has been rephrased: ‘’students that were overweight or obese increased as age increased’’. The ‘’except in students’’ expunged. Line 303.

Reviewer: Line 314: Use “UI”

Response: ..admitted into UI. Line 380.

Reviewer: Line 315: Your sample categorization does not only include young adults, but older adults too.

Response: Thank you for pointing out this important fact. The sentence has been revised to read ‘’a large population of adolescents, young and older adults’’. Line 379.

Reviewer: Lines 329 to 330: This trend persists 329 till later years of postgraduate programmes depicting increasing risk of obesity at middle age and 330 later in adulthood. ( This sentence needs to be re written to be make it clearer)

Response: This sentence has been rephrased thus:

‘’We found a consistently increasing trend of overweight and obesity with age, which persists significantly in many of the study participants till the later years of their postgraduate study, suggesting an increased risk of obesity till middle age and later in adulthood’’. Line 394-396.

Reviewer: Line 338: “ without investment in the built environment for physical activity.” The meaning of this is not clear. Please clarify.

Response: We have addressed this unintended confusing statement by rephrasing the sentence as: ‘’with the lack of investment in developing an environment that promotes physical activity’’. Line 408.

Reviewer: Line 347: Please what is meant by “increasing shift”? Does this mean that under nutrition was decreasing and over nutrition was increasing as the year of entry increased? Please make your sentences clearer and easier to follow and understand. You can cite any of the figures that illustrates this and give a reason for this finding.

Response: No, undernutrition was not decreasing but rather, there was a rising trend in the underweight status of the participants as the year of entry increased. However, while undernutrition was increasing, there was an increasing shift from underweight to overnutrition (overweight/obesity) simultaneously across each year of admission to the University (table 2, figure 3). Line 416-418. The is the dual burden of malnutrition phenomenon due to nutrition and economic transitions in developing countries. Line 428-430. It’s regrettable that this information (fig 3) was inadvertently omitted initially but has been included in the revised manuscript as guided by your comment. Thank you

Reviewer: Line 343: The discussion in the section on Double Burden of Malnutrition should reflect the findings and numbers reported in the results. This is not the case presently. I will suggest a re-write of this section discussing the figures and relevance of both underweight and overweight, comparison with other reports, causes and implications to the wider society.

Response: Thank you for your comments and suggestion that have stimulated further discussion on the double burden of malnutrition. We have included more discussions on the dual burden of malnutrition, the relationship and the biological link between the diverse forms of malnutrition beyond coexistence. However, underweight was the only index of undernutrition that was explored in this study. Other indices of undernutrition – stunting, wasting and micronutrient deficiencies were not part of the outcome variables for this study and may not be sufficiently supported by our data. We did not find a significant association between underweight and chronic disease risks. However, we presented the possible complications and public health implications of undernutrition and the need for more investigations in future studies. Line 434-453. 

Reviewer: 403 : change tiny to the exact percentage>

Response: This study shows that only 5.6% of the study participants had Electrocardiography. Line 471.

Reviewer: 433 : “Young people” should be changed to “ Adolescents and young adults” all over the manuscript as has been used previously in your manuscript.

Response: This comment is well noted and has been addressed where appropriate. Thank you.

Reviewer: Though this manuscript’s title focuses on Obesity, much more ought to have been said about under weight (under nutrition )

Response: We have included more discussions on underweight as stated in line 437-453.

Reviewer: Interventions : This will include education of the harmful effect of poor nutrition and advice and on the benefits of healthy eating including components of a balanced diet. The positive effect of role models. Deterrents for higher energy drinks and policy statements such as increased tax on such high energy drinks etc

Response: These points are well appreciated and are part of our discussion. Line 524-528, 544-547. 

Reviewer: Though i could find the figures, there were no tables present for inspection.

Response: We regret the error that occurred, the tables have been included in the revised manuscript. 

Reviewer: A major flaw of this manuscript is that the author’s posture and discussion is focused on the students and their learning environment (UI). This manuscript gives the impression or idea that these are students who have been in UI. However, this is not the case. The sample used are fresh students who have been out of the school and are just resuming into UI. Therefore, this sample is not affected or do not have any impact yet by the school environment. The BMI measurements were taken at the point of admission into the school and not during their study. Therefore, this does not reflect the impact of UI environment as the authors have made it appear. This should be corrected all over the manuscript.

Response: Thank you for this comment. It reflects how much more we need to do to communicate some peculiar aspects of the study population and environment. First, some of the postgraduate students were former UI undergraduates who had returned for postgraduate studies and had their medical screening repeated. Like many institutions, UI gives admission preference to their young students with good academic records for postgraduate studies before considering students from other institutions. Whatever it was, this category of students may have been impacted by the UI school environment. Second, the remaining postgraduate students too were not just coming from secondary school but other Nigerian tertiary institutions. Wherever they came from, the impact of their school environment on this other category of postgraduate students cannot be ignored because it couldn’t have been zero. Overall, the postgraduate students consisted of 48.1% of the study population. 

Regarding the comment on the potential obesogenic environment in UI, the significant association between overweight/obesity prevalence and older age and postgraduate study gives room for some cautious inferences. First, it bears repeating that some of the study participants who enrolled for postgraduate study were returning undergraduate students of UI who had their medical screening repeated for postgraduate admission during the period under review. Unfortunately, we couldn’t synthesize this information from our study because the data was fully anonymized. Second, the result of some studies to date show that there has not been any form of intervention for adolescents and young adults with high BMI in universities and the general population in Nigeria and Africa. Any institution where there is no decisive policy and interventions to tackle obesity could be defined as obesogenic. Moreover, studies show that university students lack knowledge about healthy food choices. Therefore, both undergraduate students and postgraduate students admitted through UI or from tertiary institutions other than UI who were previously underweight or overweight may naturally follow the life course of the obesity trajectory, all things being equal. This is one plausible way to explain the consistent and significant rise of overweight/obesity across the age groups and during the postgraduate years. Of course, this and others need further study.

Consequently, we have made another attempt at revising the manuscript to reflect the peculiar characteristics of the study environment and participants in our discussion. Line 157-165, 396-412. Thank you very much. 

Reviewer: The other issue is that the manuscript is much too long and almost boring. The information can be given in fewer words and in clear sentences too. The authors should engage an English language editor to help make their work clearer and more focused.

Response: Thank you. This point has been addressed in your previous comment on the introduction section and also in the body of the manuscript. 

Reviewer: I will also suggest that the tables be included in the submission.

Response: This point is well noted and has been addressed in the revised manuscript. Thank you.

---

## [Decision Letter · Decision Letter 1]

17 Oct 2022

PONE-D-22-08089R1’Temporal Trends in Obesity and Chronic disease risks Among Adolescents and Young Adults: A Ten-year Review at a Tertiary institution in Nigeria'PLOS ONE

Dear Dr. Oluwasanu,

Thank you for submitting your manuscript to PLOS ONE. After careful consideration, we feel that it has merit but does not fully meet PLOS ONE’s publication criteria as it currently stands. Therefore, we invite you to submit a revised version of the manuscript that addresses the points raised during the review process. I agree with the referees and I also believe that the paper would benefit from being more concise.

We look forward to receiving your revised manuscript.

Kind regards,

Filipe Prazeres, MD, MSc, Ph.D.

Academic Editor

PLOS ONE

Journal Requirements:

Reviewers' comments:

Reviewer's Responses to Questions

**Comments to the Author**

1. If the authors have adequately addressed your comments raised in a previous round of review and you feel that this manuscript is now acceptable for publication, you may indicate that here to bypass the “Comments to the Author” section, enter your conflict of interest statement in the “Confidential to Editor” section, and submit your "Accept" recommendation.

Reviewer #1: All comments have been addressed

Reviewer #2: All comments have been addressed

2. Is the manuscript technically sound, and do the data support the conclusions?

Reviewer #1: Yes

Reviewer #2: Yes

3. Has the statistical analysis been performed appropriately and rigorously? 

Reviewer #1: Yes

Reviewer #2: Yes

4. Have the authors made all data underlying the findings in their manuscript fully available?

Reviewer #1: Yes

Reviewer #2: Yes

5. Is the manuscript presented in an intelligible fashion and written in standard English?

Reviewer #1: Yes

Reviewer #2: Yes

6. Review Comments to the Author

Reviewer #1: The suggested revisions have been accepted or manuscript edited for those suggestions. Paper would benefit from being concise.

Reviewer #2: Dear Authors, I congratulate you on the efforts you have put into this manuscript. It is improved.

I only have the following observations

1. The introduction can still be made even more concise.

2. Line 150: The University commence in 2011….. ( Which University is referred to here, UI or PAU?)

3. Line 230: …….between 2009 and 2017 (Based on Figure 1, there was a decrease in 2018 compared to the 2016 and 2017. I therefor recommend stating between 2009 and 2017 )

7. PLOS authors have the option to publish the peer review history of their article (what does this mean?). If published, this will include your full peer review and any attached files.

Reviewer #1: **Yes: **Bright Thilagar MD

Reviewer #2: **Yes: **OGUGUA NDUBUISI OKONKWO

---

## [Author Response · Author response to Decision Letter 1]

15 Dec 2022

Journal Requirements:

Author's Response: Thank you for your review. The reference list has been revised and updated to reflect the changes in the manuscript text which was overhauled to make it more concise as variously suggested. The changes to the whole reference list are as tracked in the file labeled “Revised Manuscript with Tracked Changes”

Review Comments to the Author

Reviewer #1: The suggested revisions have been accepted or manuscript edited for those suggestions. Paper would benefit from being concise.

Author’s Response: Thank you for your painstaking review and practical suggestions so far. The manuscript has undergone further review to make it more concise and improve engagement. Some sentences and paragraphs were modified or totally expunged. The modifications and changes in the following areas are noteworthy:

Introduction, line 68 in the revised manuscript

The second paragraph of this latest revision ends in line 93 with the expunging of “This dual burden of malnutrition places heavy tolls on individuals, families, economies, and healthcare systems [22,23,24]” in lines 93 to 94 in the manuscript with tracked changes.

The third paragraph starting from line 95 was also reviewed and reduced by expunging (lines 102 to 105 in the manuscript with tracked changes), “For example, there is a preference for energy-dense foods, higher-fat intake, and more sugar-containing drinks. Reduced energy expenditure due to insufficient physical activity and sedentary activities such as long hours of watching television or other screen devices are also contributory factors in an increasingly urbanised and digitalized world.”

The fourth paragraph begins in line 106 with “The period of stay…”. Lines 111 to 112 are expunged in the manuscript with tracked changes. Also expunged is a statement in line 122, “It may as well be that universities are unwittingly sustaining an obesogenic environment critical to amplifying the biological vulnerability of young people to obesity.”

Discussion

The burden of overweight and obesity, line 377 in the revised manuscript

The second paragraph, line 395 in the manuscript with tracked changes was modified and rearranged. The first sentence was modified as, “We found a consistently increasing trend of overweight and obesity with age, which persists significantly in the consecutive admission cohorts, suggesting an increased……” This was followed by a rearrangement of the paragraph after expunging a large chunk of many sentences/statements in lines 408 to 415 in the tracked manuscript.

Double Burden of Malnutrition, line 400 in the revised manuscript

The first and second paragraphs in this sub-heading were reviewed and redundant statements were removed, lines 428 to 433 in the first paragraph and lines 435 to 437 and 441 to 443 in the manuscript with tracked changes. 

Obesity-related health conditions, line 430 in the revised manuscript

The second paragraph in this sub-heading starting in line 441 (472 in the manuscript with tracked changes) was rearranged/re-paraphrased with the removal of some statements.

Reviewer #2: Dear Authors, I congratulate you on the efforts you have put into this manuscript. It is improved. I only have the following observations:

1. The introduction can still be made even more concise.

Author’s Response: Thank you so much for your review, suggestions, and kind words. The introduction and some other sections of the manuscript have been revised, modified, re-paraphrased and some statements expunged as follows:

Introduction, line 68 in the revised manuscript

The second paragraph of this latest revision ends in line 93 with the expunging of “This dual burden of malnutrition places heavy tolls on individuals, families, economies, and healthcare systems [22,23,24]” in lines 93 to 94 in the manuscript with tracked changes.

The third paragraph starting from line 95 was also reviewed and reduced by expunging (lines 102 to 105 in the manuscript with tracked changes), “For example, there is a preference for energy-dense foods, higher-fat intake, and more sugar-containing drinks. Reduced energy expenditure due to insufficient physical activity and sedentary activities such as long hours of watching television or other screen devices are also contributory factors in an increasingly urbanised and digitalized world.”

The fourth paragraph begins in line 106 with “The period of stay…”. Lines 111 to 112 are expunged in the manuscript with tracked changes. Also expunged is a statement in line 122, “It may as well be that universities are unwittingly sustaining an obesogenic environment critical to amplifying the biological vulnerability of young people to obesity.”

Discussion

The burden of overweight and obesity, line 377 in the revised manuscript

The second paragraph, line 395 in the manuscript with tracked changes was modified and rearranged. The first sentence was modified as, “We found a consistently increasing trend of overweight and obesity with age, which persists significantly in the consecutive admission cohorts, suggesting an increased……” This was followed by a rearrangement of the paragraph after expunging a large chunk of many sentences/statements in lines 408 to 415 in the tracked manuscript.

Double Burden of Malnutrition, line 400 in the revised manuscript

The first and second paragraphs in this sub-heading were reviewed and redundant statements were removed, lines 428 to 433 in the first paragraph and lines 435 to 437 and 441 to 443 in the manuscript with tracked changes. 

Obesity-related health conditions, line 430 in the revised manuscript

The second paragraph in this sub-heading starting in line 441 (472 in the manuscript with tracked changes) was rearranged/re-paraphrased with the removal of some statements.

2. Line 150: The University commenced in 2011….. ( Which University is referred to here, UI or PAU?)

Author’s response: The statement here refers to PAU and this has been corrected in line 142 in the revised manuscript.

Thank you.

3. Line 230: …….between 2009 and 2017 (Based on Figure 1, there was a decrease in 2018 compared to the 2016 and 2017. I therefore recommend stating between 2009 and 2017)

Author’s Response: This correction has been done. Line 222 in the revised manuscript. 

Thank you.

Other significant modifications

Line 218 in the manuscript with tracked changes – Students aged 16-20 years constituted at least a quarter of the study population. This statement was changed to: Students aged 16-20 years constituted over a third of the study population, line 209 in the revised manuscript.

Line 567 in the manuscript with tracked changes – Furthermore, the study spans a ten-year period which shows trends in obesity and hypertension, thus increasing the validity and power of this study. This statement was changed to: Furthermore, the study spans a ten-year period showing the trajectory of obesity and hypertension prevalence.

Thank you.

---

## [Decision Letter · Decision Letter 2]

23 Jan 2023

PONE-D-22-08089R2Temporal Trends in Obesity and Chronic disease risks Among Adolescents and Young Adults: A Ten-year Review at a Tertiary Institution in NigeriaPLOS ONE

Dear Dr. Oluwasanu,

Thank you for submitting your manuscript to PLOS ONE. After careful consideration, we feel that it has merit but does not fully meet PLOS ONE’s publication criteria as it currently stands. Therefore, we invite you to submit a revised version of the manuscript that addresses the points raised during the review process.

We look forward to receiving your revised manuscript.

Kind regards,

Haris Khurram

Academic Editor

PLOS ONE

Journal Requirements:

Additional Editor Comments:

Dear Author(s), we understand that peer review already took a long time, but I think it is necessary to enhance the quality of a manuscript that meets the standard criteria. After thorough reading, I have suggested some changes that will definitely improve the quality of the manuscript.

In data analysis, the author(s) mentioned, “Trends in obesity and hypertension were assessed using Chi-square for trends at a 5% significance level. Independent factors associated with obesity/overweight and hypertension were evaluated using a multivariable binary logit model. Measures of effect were reported as Odds Ratio with 95% Confidence Interval (95% CI)”. I suggest rewriting it all. Chi-square is used to measure association, not trends. In the second line, if the factors are independent then why are you finding the association? Change word “independent”. Odds ratios are also used for finding the association. Measure of effect is not a suitable world.

Table 1 provided (%) but table 2 provided n(%). I suggest presenting both.

Table 4 presents chi-square, but the proper notation of chi instead of writing X

“Table 5: Multivariate analysis showing the association between socio-demographic 312 characteristics of students and Overweight and Obesity” should be, “… of students with

I am wondering to see that all p-values in tables 5 and 7 are exactly 0.001.

Table 6, for pre-hypertension, there is only a yes category in the header.

Table 7 has a different font size.

Figure 3 has a title “chart title”

Figure 5 has a dotted trend line, which is not appropriate for bar graphs and each of the year categories has the repeated word “year.”

Figure 1 and figure 2 are both combined in figure 3 so I think no need for figures 1 and 2.

In Figures 4 and 2, the author(s) presents the results for underweight, overweight, and obese. But the table only compares overweight and obese as yes or no. Why did the author(s) not measure the association with underweight, normal, overweight, and obese instead of using combined overweight as yes/no? Author(s) discussed the overweight and obese together, although the paper title has the obese only. While WHO highly encourages to deal overweight and obese as different categories.

Reviewers' comments:

Reviewer's Responses to Questions

**Comments to the Author**

1. If the authors have adequately addressed your comments raised in a previous round of review and you feel that this manuscript is now acceptable for publication, you may indicate that here to bypass the “Comments to the Author” section, enter your conflict of interest statement in the “Confidential to Editor” section, and submit your "Accept" recommendation.

Reviewer #1: All comments have been addressed

Reviewer #2: (No Response)

2. Is the manuscript technically sound, and do the data support the conclusions?

Reviewer #1: Yes

Reviewer #2: Yes

3. Has the statistical analysis been performed appropriately and rigorously? 

Reviewer #1: Yes

Reviewer #2: Yes

4. Have the authors made all data underlying the findings in their manuscript fully available?

Reviewer #1: Yes

Reviewer #2: Yes

5. Is the manuscript presented in an intelligible fashion and written in standard English?

Reviewer #1: Yes

Reviewer #2: Yes

6. Review Comments to the Author

Reviewer #1: Reviewer comments have been addressed to satisfaction. Errors to sentence structure has been changed. The introduction is more concise.

Reviewer #2: Dear Author, your work is much improved.

However, i feel that in line with the title of your paper, suggesting that the trends are being examined, table 2 clearly suggests an increase in underweight ( the absolute numbers and percentages of underweight for the first five years of the study period are less than for the later five years ) and a decrease in both overweight and obesity (the absolute numbers and percentages of both over weight and obesity for the first five years are more than for the last five years). This ought to be clearly mentioned and discussed .

Also this appears to contrary to the statement in line 226 and 227 "On the other hand, there was an increasing shift from underweight to overnutrition (overweight and obesity) across each year of admission into the University..

Though cancer is mentioned as one of the variables sought after in the records in line 185, this is not reported on in the results. Please state the number of cancer diagnosed and if none then mention it.

7. PLOS authors have the option to publish the peer review history of their article (what does this mean?). If published, this will include your full peer review and any attached files.

Reviewer #1: No

Reviewer #2: **Yes: **OGUGUA NDUBUISI OKONKWO

---

## [Author Response · Author response to Decision Letter 2]

22 Feb 2023

Additional Editor Comments:

In data analysis, the author(s) mentioned, “Trends in obesity and hypertension were assessed using Chi-square for trends at a 5% significance level. Independent factors associated with obesity/overweight and hypertension were evaluated using a multivariable binary logit model. Measures of effect were reported as Odds Ratio with 95% Confidence Interval (95% CI)”. I suggest rewriting it all. Chi-square is used to measure association, not trends. In the second line, if the factors are independent then why are you finding the association? Change word “independent”. Odds ratios are also used for finding the association. Measure of effect is not a suitable world.

Response. Thank you for your observation and suggestions. 

Line 197, independent removed

Line 198, effect removed, association inserted.

The section has been revised accordingly to read:

The prevalence of overweight and obesity and the trends during the period in review were determined. The association between the students’ sociodemographic characteristics, hypertension and overweight/obesity was assessed using Chi-square at a 5% significance level. The factors associated with obesity/overweight and hypertension were evaluated using a multivariable binary logit model. Measures of association were reported as Odds Ratio with a 95% Confidence Interval (95% CI).

Editor. Table 1 provided (%) but table 2 provided n(%). I suggest presenting both. 

Response: n(%) was provided in table 1, line 233

Line 238, fig 1 cancelled, table 2 inserted

Editor. Table 4 presents chi-square, but the proper notation of chi instead of writing X 

Response: Chi-square proper notation inserted in line 197 and used in Tables 4 and 6. Thank you.

Editor. “Table 5: Multivariate analysis showing the association between socio-demographic 312 characteristics of students and Overweight and Obesity” should be, “… of students with

I am wondering to see that all p-values in tables 5 and 7 are exactly 0.001.

Response: Those were the exact p-values obtained. This is not unusual given the very large sample size that we had. 

The title for table 5 has been revised, ‘with’ substituted for ‘and’ line 312

Editor. Table 6, for pre-hypertension, there is only a yes category in the header.

Response. A ‘no’ category was added, line 327. Thank you.

Editor. Table 7 has a different font size.

Response. The font size was increased and aligned with other tables.

Editor. Figure 3 has a title “chart title”

Response. The “chart title” was removed and figure 3 converted to figure 1.

Editor. Figure 5 has a dotted trend line, which is not appropriate for bar graphs and each of the year categories has the repeated word “year.”

Response. Figure 5 was revised accordingly but completely removed because it didn’t provide any additional information or value more than what the tables provided. Thank you.

Editor. Figure 1 and figure 2 are both combined in figure 3 so I think no need for figures 1 and 2.

Response. This is very true, figures 1 and 2 were removed. Thank you.

Editor. In Figures 4 and 2, the author(s) presents the results for underweight, overweight, and obese. But the table only compares overweight and obese as yes or no. Why did the author(s) not measure the association with underweight, normal, overweight, and obese instead of using combined overweight as yes/no? Author(s) discussed the overweight and obese together, although the paper title has the obese only. While WHO highly encourages to deal overweight and obese as different categories.

Response. Undernutrition and its associations have historically received more public health attention than over-nutrition in sub-Saharan Africa, especially in Nigeria. Also, there are intervention studies and programmes on undernutrition, including the ongoing national school feeding programme by the government of Nigeria. However, overnutrition is a growing, silent epidemic among children and young people with scarce data and without attention and intervention yet. Besides, our study is not just about the prevalence of overweight or obesity but its associations with chronic disease risks. The data for our study is restrictive as it lacks adiposity measures beyond BMI and cannot support robust research analysis to generate (new) findings on the various forms and manifestations of undernutrition and chronic disease risk associations. Hence, the objectives of our study focused mainly on overnutrition based on BMI (the only available indicator of adiposity), and we did not intend to repeat research on undernutrition or seek associations between underweight and chronic disease risks which could not be supported by our data. Even for the normal category, without additional adiposity measures for proper categorization, it may yield misleading results. These were the reasons why we did not include underweight and normal in the table. Figures 4 and 2 were used suitably for the purpose of comparison only. 

Combined category

First, the combined category of “overweight and obese” refers to a BMI of 25 or higher. Overweight and obesity are defined as abnormal or excessive fat accumulation that may impair health. The combined category is preferably used to avoid underestimation of measures when the value (absolute number or percentage) of the separate categories is low as we have in our data. Compared with the overall prevalence, there was a marked decrease in the absolute numbers and percentages of the separate categories of overweight and obese, corresponding with the lower postgraduate admission rate during some years in the period in review. The obese category is most significantly reduced with the lowest rate of postgraduate admission, especially in 2017. Second, this combination is also expedient in our study since there are no additional measures of adiposity (information on fat location/distribution) that can be used to estimate the risk for associated NCDs except for high BMI. The key difference between being overweight and obese lies in the extent of accumulation of excessive body fats and this does not give information about its location/distribution. Except for those with an exceeding large total fat mass (BMI), where fat is accumulated/distributed is much more important than the extent (amount) of the separate BMI categories in the prediction of comorbidities such as cardiovascular disease, diabetes, hypertension, malignancies, or overall death rates. This is one of the well-documented limitations of BMI as an indicator of body fat and associated chronic disease risk. 

Obesity, defined as a disease or a medical condition that poses risk to health, is a generic term that includes overweight and obese and/or other categories of obesity. Obese or overweight are separate categories of obesity defined by BMI only (anthropometric obesity). Also, Obesity is categorized not only by BMI but other measures like waist circumference, waist-hip ratio (central or abdominal obesity). Obesity includes individuals with normal BMI but are obese based on these other measures. So, the term obesity in our title is not restricted to obese only but covers both overweight and obese. However, based on your comment, we added overweight to the title of our study, to read and be interpreted as, the temporal trends in overweight (as a distinct/separate BMI category) and obesity (as a combined BMI category of overweight and obese)…. Thank you. 

Reviewer #2: 

Dear Author, your work is much improved.

However, i feel that in line with the title of your paper, suggesting that the trends are being examined, table 2 clearly suggests an increase in underweight ( the absolute numbers and percentages of underweight for the first five years of the study period are less than for the later five years ) and a decrease in both overweight and obesity (the absolute numbers and percentages of both over weight and obesity for the first five years are more than for the last five years). This ought to be clearly mentioned and discussed.

Also this appears to contrary to the statement in line 226 and 227 "On the other hand, there was an increasing shift from underweight to overnutrition (overweight and obesity) across each year of admission into the University.

Response. Truly, when compared with the overall percentage and number, there is a marked decrease in the absolute numbers and percentages of both overweight and obese in 2014, 2016 and 2017 (now included in line 242) corresponding with the lower postgraduate admission rate during these years. Despite the obvious dip in the values of overweight and obesity during these years, the prevalence of overweight was consistently higher than underweight, except for 2017 (Table 2, line 243). The validity of the statement about the transition from undernutrition to overnutrition across each admission year is sustainable (fig 1). It is difficult and beyond the scope of our study to adduce any reasons for the change in either the admission or medical registration pattern or discuss its implications differently. However, “an increasing” shift from underweight to overnutrition was changed to a shift from underweight to overnutrition (line 244, 407).

Thank you for all your comments and valuable suggestions.

Reviewer #2. Though cancer is mentioned as one of the variables sought after in the records in line 185, this is not reported on in the results. Please state the number of cancer diagnosed and if none then mention it.

Response. There was no record of cancer diagnosis. This statement has been added in line 266.

Thank you.

General corrections:

Title: overweight was added as explained in the Editor’s comment above.

Abstract

line 58 obesity (10.4%) changed to obese (10.4%)

line 63 ‘rising trends in prevalence’ was cancelled and inserted, a higher prevalence of overweight and obesity than underweight among the participants.

Thank you.

---

## [Editor Report · Decision Letter 3]

6 Mar 2023

Temporal Trends in Overweight and Obesity and Chronic disease risks Among Adolescents and Young Adults: A Ten-year Review at a Tertiary Institution in Nigeria

PONE-D-22-08089R3

Dear Dr. Oluwasanu,

We’re pleased to inform you that your manuscript has been judged scientifically suitable for publication and will be formally accepted for publication once it meets all outstanding technical requirements.

Kind regards,

Haris Khurram

Academic Editor

PLOS ONE
---

## [Editor Report · Acceptance letter]

13 Mar 2023

PONE-D-22-08089R3 

Temporal Trends in Overweight and Obesity and Chronic disease risks Among Adolescents and Young Adults: A Ten-year Review at a Tertiary Institution in Nigeria 

Dear Dr. Oluwasanu:

I'm pleased to inform you that your manuscript has been deemed suitable for publication in PLOS ONE. Congratulations! Your manuscript is now with our production department. 

Kind regards, 

on behalf of

Dr Haris Khurram 

Academic Editor

PLOS ONE